# Open-Qwen2VL: Compute-Efficient Pre-Training of Fully-Open Multimodal LLMs on Academic Resources

**Weizhi Wang**[1], **Yu Tian**[2], **Linjie Yang**[2], **Heng Wang**[3], **Xifeng Yan**[1]
[1]UC Santa Barbara, [2]Seed Vision Team, ByteDance, [3]Nvidia Research
`weizhiwang@ucsb.edu`

## Abstract

The reproduction of state-of-the-art multimodal LLM pre-training faces barriers at every stage of the pipeline, including high-quality data filtering, multimodal data mixture strategies, sequence packing techniques, and training frameworks. We introduce Open-Qwen2VL, a fully open-source 2B-parameter Multimodal Large Language Model pre-trained efficiently on 29M image-text pairs using only 220 A100-40G GPU hours. Our approach employs low-to-high dynamic image resolution and multimodal sequence packing to significantly enhance pre-training efficiency. The training dataset was carefully curated using both MLLM-based filtering techniques (e.g., MLM-Filter) and conventional CLIP-based filtering methods, substantially improving data quality and training efficiency.

The *Open-Qwen2VL* pre-training is conducted on academic level 8xA100-40G GPUs at UCSB on 5B packed multimodal tokens, which is 0.36% of 1.4T multimodal pre-training tokens of Qwen2-VL. The final instruction-tuned *Open-Qwen2VL* outperforms partially-open state-of-the-art MLLM Qwen2-VL-2B on various multimodal benchmarks of MMBench, SEED-Bench, MMstar, and MathVista, indicating the remarkable training efficiency of *Open-Qwen2VL*.

We open-source all aspects of our work, including compute-efficient and data-efficient training details, data filtering methods, sequence packing scripts, pre-training data in WebDataset format, FSDP-based training codebase, and both base and instruction-tuned model checkpoints. We redefine "fully open" for multimodal LLMs as the complete release of: 1) the training codebase, 2) detailed data filtering techniques, and 3) all pre-training and supervised fine-tuning data used to develop the model.

| | | |
|---|---|---|
| 🌐 | **Website** | `https://victorwz.github.io/Open-Qwen2VL` |
| ⭘ | **Code** | `https://github.com/Victorwz/Open-Qwen2VL` |
| 🤗 | **Models** | `https://huggingface.co/weizhiwang/Open-Qwen2VL` |
| 🤗 | **Data** | `https://huggingface.co/datasets/weizhiwang/Open-Qwen2VL-Data` |

## 1 Introduction

The Multimodal Large Language Models (MLLMs) (Anthropic, 2025; OpenAI, 2023; Chen et al., 2024b; Wang et al., 2024a; Wu et al., 2024) present strong emergent capabilities on multimodal understanding and visual reasoning, eliciting the Artificial Intelligence Applications to comprehend and analyze images, charts, and PDF documents. Different from conventional Vision-Language Models (VLMs) (Radford et al., 2021; Jia et al., 2021), trained on image-text caption data from scratch with small model size, the MLLMs are typically constructed on a well-trained text-only LLM and then continually pre-trained on diverse large-scale multimodal data. However, recent state-of-the-art MLLMs are neither "fully-open" to the community for reproduction nor compute-friendly to academic institutions with limited GPUs. In Table 1, we compare the openness of recent SOTA MLLMs of VILA (Lin et al., 2024), MM1 (Zhang et al., 2024), Ideflics (Laurençon et al., 2024), BLIP-3 (Xue et al., 2024), Llama-3.2-Vision (Dubey et al., 2024), Phi-3.5-Vision (Abdin et al., 2024),

| Models | Data Filtering Techniques | Sequence Packing Scripts | Pre-Training Data | Pre-Training Codebase | Base Model Checkpoint | SFT Data | Instruct Model Checkpoint |
|---|---|---|---|---|---|---|---|
| VILA | None | None | Open | Open | Open | Open | Open |
| MM1 | Closed | Closed | Closed | Closed | Closed | Closed | Closed |
| Ideflics | Open | Open | Open | Open | Open | Open | Open |
| BLIP-3 | Closed | Closed | Open | Closed | Open | Closed | Open |
| Llama-3.2-Vision | Closed | Closed | Closed | Closed | Open | Closed | Open |
| Phi-3.5-Vision | Closed | Closed | Closed | Closed | Closed | Closed | Open |
| Qwen2VL | Closed | Closed | Closed | Closed | Open | Closed | Open |
| *Open-Qwen2VL* | Open | Open | Open | Open | Open | Open | Open |

Table 1: Comparisons of openness between several state-of-the-art MLLMs.

and Qwen2VL (Wang et al., 2024a). Even if most of the SOTA MLLMs release their base or instruction-tuned model checkpoints, their killer secrets of data filtering techniques, sequence packing scripts, pre-training data, training codebase, etc are completely close-source, in which they even hide such technical details in their technical reports.

In this work, we introduce *Open-Qwen2VL*, a 2B-parameter MLLM which outperforms close-source Qwen2-VL-2B on various multimodal benchmarks and achieves outstanding compute efficiency. *Open-Qwen2VL* is pre-trained on approximately 5B well-curated high-quality caption data tokens, which is 0.36% of 1.4T multimodal tokens of Qwen2-VL (Wang et al., 2024a) pre-training. Such remarkable data-efficiency enables us to perform the pre-training on academic-level computing resources of 8*A100-40G GPUs. In addition, we conduct compressive visual projector (Yao et al., 2024) to scale-down 729 image patches to 144 visual tokens and perform multimodal sequence packing to further enhance the pre-training efficiency.

We perform comprehensive ablation studies on pre-training data mixture strategies and data filtering models. The best pre-training data consists of CC3M-CC12M-SBU (Ordonez et al., 2011; Sharma et al., 2018a) caption dataset curated by CLIP and DataComp-Medium caption dataset curated by both the DFN-CLIP and MLM-Filter. Adopting efficient MLLM as the data filtering model significantly enhances the model capabilities on various benchmarks. Additionally, we scale up the visual supervised fine-tuning (SFT) data to 10M level (Guo et al., 2024) to further enhance the model capabilities.

We open-source everything to the community to help easy and convenient reproductions to our model *Open-Qwen2VL*, including the data filtering details, sequence packing scripts, pre-training data in webdataset format, training codebase based on FSDP, and both base model and instruction-tuned model checkpoints. Meanwhile, our open-source codebase is the first comprehensive solution that supports all stages of Multimodal Large Language Model training, including large-scale caption data preparation, quality score generation, data filtering, multimodal sequence packing, pre-training, supervised fine-tuning, and evaluations on multimodal benchmarks. We redefine "fully open" for multimodal LLMs as the complete release of: 1) the training codebase, 2) detailed data filtering techniques, and 3) all pre-training and supervised fine-tuning data used to develop the model. We wish to demonstrate that the research on pre-training is not only a game for giant tech companies and encourage the academic community to work on pre-training data and pipeline research even with very limited computing resources.

## 2 Compute-Efficient Multimodal Pre-Training

### 2.1 Dataset Choices and High-Quality Data Filtering

The current advanced multimodal LLMs are continually pre-trained on a large-scale image-text caption dataset. In addition to image-text caption dataset, some of latest MLLMs like VILA(Lin et al., 2024), MM1(McKinzie et al., 2024), DeepSeek-VL2 (Wu et al., 2024) also mix the image-text interleaved data with caption data for multimodal pre-training. Mixing image-text caption data and interleaved data will enhance the multimodal in-context

| ID | Dataset | Filtering Model | #Image-Text Pairs | Resources |
|----|---------|-----------------|-------------------|-----------|
| 1 | CCS | CLIP | 8.5M | https://github.com/salesforce/BLIP |
| 2 | DataComp | DFN | 15M | huggingface:apf1/datafilteringnetworks_2b |
| 3 | LAION | CLIP | 15M | https://github.com/salesforce/BLIP |
| 4 | DataComp | MLMFilter & DFN | 19.9M | huggingface:weizhiwang/Open-Qwen2VL-Data |

Table 2: Image-Text Caption Datasets for *Open-Qwen2VL* pre-training.

learning and multi-image reasoning capabilities of MLLMs. However, MM1 (McKinzie et al., 2024) demonstrates introducing image-text interleaved documents into pre-training data will reduce the zero-shot single-image reasoning and understanding capabilities of base MLLMs. Thus, to control the scale of pre-training data and ensure the pre-training efficiency, *Open-Qwen2VL* focuses on the pre-training paradigm on image-text caption data only.

To motivate the easy reproduction to our work from the community, we choose 4 most popular image-text caption datasets shown in Table 2, which are widely used in open-source vision-language model pre-training. BLIP-1 (Li et al., 2022) releases the high-quality caption data curated from a combination of CC3M-CC12M-SBU (CCS) using CLIP-based (Radford et al., 2021; Hessel et al., 2021) filtering. LAION-400M (Schuhmann et al., 2021) implements a strict 0.3 threshold based on CLIP image-text cosine similarity to curate its high-quality dataset of 400M image-text caption pairs. We download the CCS and LAION caption datasets based on the release image-urls using img2dataset (Beaumont, 2021) tool. We only download 15M LAION data to perform controlled-size data mixture ablation studies in Section 2.5.

Secondly, we choose DataComp-Medium-128M (Gadre et al., 2023) as another pre-training data choice. Based on the leaderboard of DataComp medium filtering track performance, Data-Filtering-Network (DFN) (Fang et al., 2023) is the top-1 independent data filter on the leaderboard[1]. We successfully download 99.8M out of 128M original released 128M DataComp-Medium data. Then we adopt the official resharder script[2] to select the DFN-curated high-quality subset based on the released top-15% data uids from DFN[3]. It is worth noting that the DFN only releases the top-15% curated data rather than the DFN model checkpoint. Thus, it is impossible to change the retained data fraction based on the quality scores generated by DFN-model. For DataComp-DFN high-quality dataset, finally we get 15M image-text caption data.

The MLLM-based data filtering method emerges since the introduction of MLM-Filter (Wang et al., 2024c), in which these methods adopt efficient MLLM as the high-quality data filter instead of CLIP model. MLM-Filter provides four distinct image-text data quality metric for high-quality data filtering, including image-text matching (ITM), object detail fulfillment (ODF), caption text quality (CTQ), and semantic understanding (SU). Based on the conclusions from ATIQE (Huang et al., 2024), the Semantic Understanding (SU) quality metric yields the best performance for MLLMs trained on the high-quality data curated from such metric. Thus, we generate the SU quality scores for DataComp-Medium data using `mlm-filter-qwen2.5-1.5b-gpt4o`[4] data filtering model and set filtering score threshold as 85 out of 100. With such threshold, We get 8M MLM-Filter curated data and union them with DFN-15M. After deduplication, we get 19.9M high-quality data.

---

[1]https://www.datacomp.ai/dcclip/leaderboard.html

[2]https://github.com/mlfoundations/datacomp/blob/main/resharder.py

[3]https://huggingface.co/datasets/apf1/datafilteringnetworks_2b/blob/main/datacomp_medium_dfn_20m_inds.npy

[4]https://huggingface.co/weizhiwang/mlm-filter-qwen2.5-1.5b-gpt4o

## 2.2 Model Architecture with Low-to-High Image Resolution

We adopt a simple architecture with Qwen2.5-1.5B-Instruct LLM Backbone (Team, 2024), Adaptive Average-Pooling Visual Projector (Yao et al., 2024), and SigLIP-SO-400M Vision Encoder (Zhai et al., 2023). Specifically, the Adaptive Average-Pooling Visual Projector contains an Adaptive Average-Pooling layer followed by a two-layer MLP. With the Adaptive Average-Pooling layer, we can scale the 729 output visual patches from SigLIP to any resolution. We adopt 144 visual tokens for representing an image in the pre-training stage and scale up the resolution to vanilla 729 visual tokens in SFT stage. Such low-to-high image resolution significantly enhances the MLLM pre-training efficiency and does not hurt the high-resolution image understanding of the final MLLM after SFT stage.

*Open-Qwen2VL* does not adopt advanced designs of 2d-Multimodal RoPE (Wang et al., 2024a) and naive dynamic resolution (Wang et al., 2024a) to save computes and ensure the training efficiency. Moreover, for academic computing resources, downloading images and saving in original resolution require huge disk space, which is unavailable in most of academic institutions. During our data downloading process with `img2dataset` (Beaumont, 2021), we resize the smaller side of the image to 512 pixels and keep the aspect ratio, which makes us not able to adopt naive dynamic resolution in the pre-training stage.

For both the pre-training and SFT stages, we freeze the parameters of vision encoder and make the parameters of projector and LLM backbone trainable to save more computes. However, recent studies (Wang et al., 2024a; Zhang et al., 2024) demonstrate that making the vision encoder trainable can further enhance the visual understanding capabilities of the MLLMs. We leave it as an ablation study for investigations.

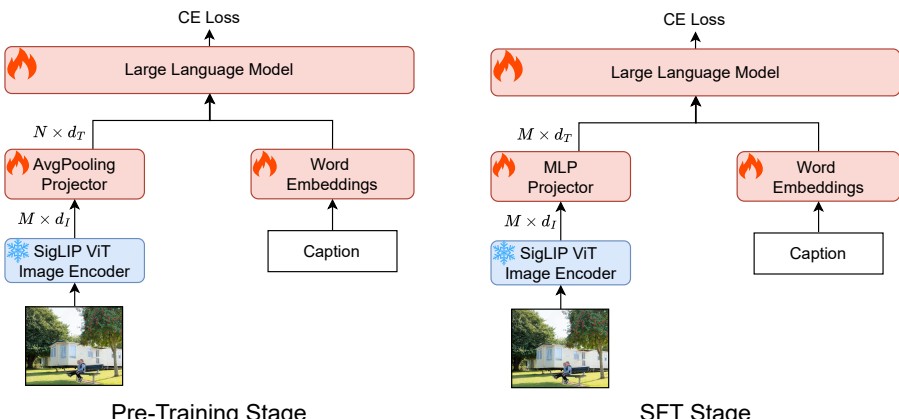

Figure 1: Model Architecture of *Open-Qwen2VL*. Here $M = 729$, $N = 144$ are the number of image patch tokens and number of projected visual tokens during the pre-training stage, respectively.

## 2.3 Multimodal Sequence Packing

Because the large-scale image-text data are varied in its length, simply batchfying a set of examples based on similar length and padding them to the longest sequence will lead to a large portion of padding tokens in each training batch. Such a large amount of padding tokens will result in heavy compute-waste and training inefficiency. Thus, we introduce multimodal sequence packing to regroup the image-text caption data into sequence groups closest to 4096 context length.

The algorithm for the multimodal sequence packing is presented in Algorithm 1. Since we download and pack all image-text caption data into webdataset format and each webdataset tar file contains exactly 10k image-text caption data, the proposed multimodal sequence packing intends to regroup such 10k pairs into a set of multimodal sequences with 4096 context length.

---

**Algorithm 1** Multimodal Sequence Packing

---

**Require:**
 1: $\mathcal{D}$: Set of image-text caption data
 2: $L$: Maximum context length
 3: $p$: Padding token
**Ensure:** Packed sequences within context length $L$
 4: **function** PACKSEQUENCES($\mathcal{D}, L, p$)
 5:     $I \leftarrow \varnothing$                                                    ▷ Initialize items dictionary
 6:     **for all** $d \in \mathcal{D}$ **do**
 7:         $(T_d, V_d) \leftarrow$ ProcessCaption($d$)                              ▷ Get text tokens and visual tokens
 8:         $len_d \leftarrow |T_d| + |V_d|$                                              ▷ $|V_d| = 144$
 9:         $I[d] \leftarrow len_d$
10:     **end for**
11:     $I \leftarrow$ SortByLength($I$)                                              ▷ Sort in descending order
12:     $B \leftarrow \varnothing$                                                      ▷ Initialize bins
13:     **for all** $(d, len_d) \in I$ **do**
14:         $placed \leftarrow$ false
15:         **for all** $bin \in B$ **do**
16:             **if** $\sum_{(d\prime, len\prime) \in bin} len\prime + len_d \leq L$ **then**
17:                 $bin$.append($(d, len_d)$)
18:                 $placed \leftarrow$ true
19:                 **break**
20:             **end if**
21:         **end for**
22:         **if** $\neg placed$ **then**
23:             $B$.append($\{(d, len_d)\}$)
24:         **end if**
25:     **end for**
26:     $P \leftarrow \varnothing$                                                      ▷ Initialize packed sequences
27:     **for all** $bin \in B$ **do**
28:         $T_{bin} \leftarrow \varnothing$                                              ▷ Concatenated text tokens
29:         $V_{bin} \leftarrow \varnothing$                                              ▷ Concatenated PIL images
30:         **for all** $d \in bin$ **do**
31:             $T_{bin}$.append($T_d$)
32:             $V_{bin}$.append($V_d$)
33:         **end for**
34:         **if** $|T_{bin}| < L$ **then**
35:             $T_{bin}$.pad($p, L$)                                                  ▷ Pad to context length
36:         **end if**
37:         $P$.append($(T_{bin}, V_{bin})$)
38:     **end for**
         **return** $P$
39: **end function**

---

The multimodal sequence packing involves three major steps: computing the multimodal length of each image-text caption sample, regroup data into several bins in which the total length of each bin is closest to 4096, and concatenate the input_ids vectors and pillow-format images. We adopt First-fit-decreasing (FFD) bin packing algorithm (Johnson, 1973) to pack each image-text caption data into several bins. We also follows LLaVA to insert an <image> placeholder token at the beginning of each image-text caption. We use the default end-of-sequence token, <|im_end|> token as the separator between each image caption text.

We store each packed multimodal sequence to a pickle file, because pickle support storing data in different formats like pillow-image and torch input_ids tensor in one file. Finally, each pickle file contains the following dictionary for each packed multimodal sequence:

- "images": a list of pillow image objects;

- "input_ids": torch Long Tensor with image placeholder token;

- "lengths": a list of integers to record the multimodal length of each image-text caption data.

| Benchmark | Data Mixture | | | |
| | 1 + 2 (23.5M) | 1 + 3 (23.5M) | 1 + 2 + 3 (38.5M) | 1 + 4 (28.4M) |
|---|---|---|---|---|
| *General Benchmark* | | | | |
| MMMU$_{val}$ | **38.9** | 37.2 | 36.7 | 38.0 |
| MMBench$_{dev}$ | 75.6 | 75.9 | 75.9 | **77.3** |
| SEEDBench $-$ Img$_{dev}$ | 68.9 | **69.6** | 68.9 | 68.7 |
| MMStar | 39.6 | **41.7** | **41.7** | 41.3 |
| *OCR VQA* | | | | |
| AI2D$_{test}$ | 56.3 | 55.5 | **57.3** | 56.8 |
| TextVQA$_{val}$ | 55.1 | **57.4** | **57.4** | 57.0 |
| *Math Reasoning* | | | | |
| MathVista$_{testmini}$ | 28.6 | 28.1 | **28.7** | 28.6 |
| *Hallucination* | | | | |
| POPE | 79.2 | **80.1** | 77.7 | **80.1** |
| Average | 55.3 | 55.4 | 55.5 | **56.0** |

Table 3: Benchmark performance of MLLMs pre-trained on different pre-train data mixture and fine-tuned on controlled LLaVA-665k instructions. Remarks for each dataset: 1: CCS-CLIP; 2: DataComp-DFN; 3: LAION-CLIP; 4: DataComp-MLM-Filter & DFN.

## 2.4 Training Infrastructure and Codebase

We develop our training codebase based on Prismatic-VLM (Karamcheti et al., 2024). The original codebase only supports SFT on single-image instructions and we heavily modify its dataloader and batch preparation to support the multimodal packed sequences with multiple-images in one sequence. We retain its Fully-Sharded Distributed Parallel (torch-FSDP) trainer, which we find that it significantly accelerates the training compared with LLaVA codebase using DeepSpeed-Zero3. Although FSDP and DeepSpeed-Zero3 utilize the same model sharding algorithm, our FSDP-based implementation achieves approximately 17% faster for each training step than the DeepSpeed implementation, consistent with findings reported by Karamcheti et al. (2024).

## 2.5 Ablations on the Data Mixture

After preparing and sequentially-packing the 4 image-text caption dataset, we conduct ablation studies to investigate the effects of different data mixtures to the performance of final MLLMs. Since there is 16 combinations between the four datasets, we only consider 4 combinations. The CCS-CLIP data is fixed and we incrementally add other three datasets with it. For each dataset group, we pre-train the MLLM for one epoch on packed multimodal sequences and then fine-tune the base MLLM on LLaVA-665k instruction dataset (Liu et al., 2023). The training details and hyperparameters are available in Appendix Table 8. Then we evaluate each model ablations on multimodal benchmarks of AI2D-test (Kembhavi et al., 2016), TextVQA-val (Singh et al., 2019), POPE (Li et al., 2023b), MMMU-val (Yue et al., 2024), MMBench-v1.0-dev (Liu et al., 2024), SEEDBench-imge-dev (Li et al., 2023a), MMStar (Chen et al., 2024a), and MathVista-test-mini (Lu et al., 2023).

**Results.** The benchmark results of each pre-trained and fine-tuned MLLM using different data mixtures are presented in Table 3. Since the DataComp-DFN and LAION are both web-crawled data and adopt similar CLIP-based data filtering techniques, the data mixtures of these two dataset with CCS achieves very similar model performance. Moreover, simply mixing three CCS-CLIP, DataComp-DFN, and LAION-CLIP does not achieve better performance, which might be caused by the high data homogeneity between DataComp-DFN data and LAION-CLIP data. Surprisingly, adding very small amount of high-quality data

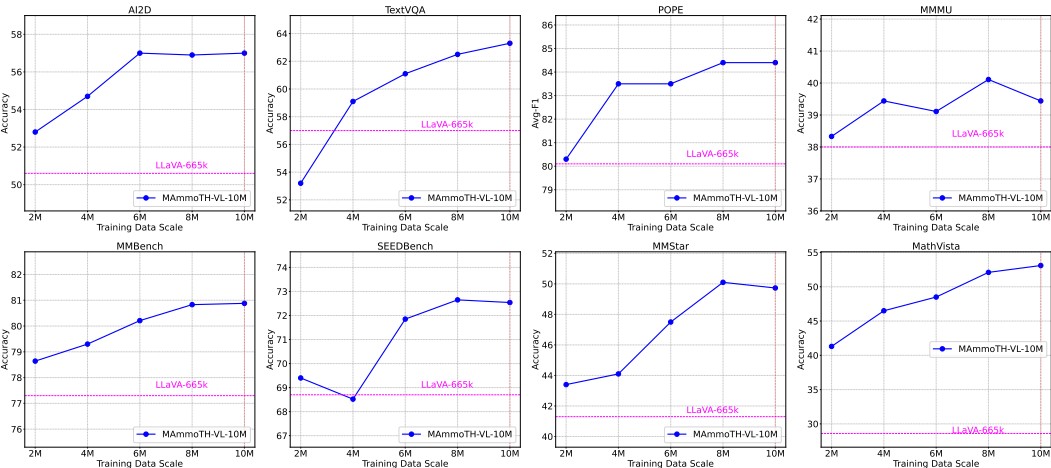

Figure 2: The scaling effects of visual SFT data from LLaVA-665k to MAmmoTH-VL-si-10M. We evaluate the checkpoints every 2M training samples.

(5M) curated by a different efficient-MLLM based data filter, MLM-Filter can significantly enhance the model performance, achieving +0.5 average performance improvements. We suppose that MLLM-based data filter may introduce a different data distribution into the pre-training set, which brings new knowledge for enhancing the MLLM capabilities.

Finally, the pre-training of *Open-Qwen2VL* on the best data mixture takes about 220 A100-40G GPU hours, and the SFT on LLaVA-665k instructions takes 48 A100-40G GPU hours.

## 3 Scaling-Up Supervised Fine-Tuning

### 3.1 SFT Dataset

After the ablation studies on the pre-training mixture, we further scale-up the visual SFT data from LLaVA-665k (Liu et al., 2023) to MAmmoTH-VL-10M (Guo et al., 2024) to further enhance the understanding and reasoning capabilities of MLLM. We only use the 10M single-image subset for visual SFT and do not include the additional LLaVA-OneVision-2M for further SFT on mixed image and video data. The MAmmoth-VL-10M data requires over 200GB CPU memory if the original LLaVA-style dataloader is adopted to load the full 10M json file data into memory in distributed multi-process. To comply with the limited CPU memory of our server, we store each data sample in the 10M full json data into single json file, and meanwhile generate a 10M-indices file for loading into the memory. Each index contains the path to the data sample json, the boolean value of text-only or image-text data, and the pre-computed image-text data length for batchfying. The SFT hyperparameters also follow Appendix Table 8.

### 3.2 Scaling Effects and Results

We save the checkpoints for every 2M sft instructions, which is 15625 steps under the batch size of 128. We illustrate the benchmark performance of each saved checkpoint in Figure 2. We can conclude that scaling-up SFT remarkably improves model performance on various multimodal benchmarks. Most of the benchmarks like POPE, MMMU, MMBench, and SEEDBench performance converges at the SFT scale of 8M instructions and do not improve for the final 2M data. The curves of TextVQA and MathVista performance vary from others, which present a steady improvement over the data scale. It might be caused by the lack of such pre-training math or OCR data in our curated pre-training caption dataset, making the visual math reasoning and text-based VQA become out-of-distribution tasks. For the general-pupose knowledge-based benchmarks of MMMU, SEEDBench, and MMStar, we even observe the slight performance degradation in the final 6M instruction tuning data.

| Models | InternVL2.5-2B-MPO | DeepSeekVL-2-Tiny | Qwen2-VL-2B-Ins | Open-Qwen2VL |
|---|---|---|---|---|
| # Pretrain Tokens | 277B | 8.1T | 1.4T | 5B |
| *General Benchmark* | | | | |
| MMMU$_{val}$ | **41.2** | 39.6 | 41.1 | 39.8 |
| MMBench$_{dev}$ | 72.5 | 68.3 | 68.8 | **80.9** |
| SEEDBench$_{dev}$ | **73.2** | 72.5 | 72.0 | 72.5 |
| MMStar | **54.3** | 49.9 | 46.3 | 49.7 |
| *OCR VQA* | | | | |
| AI2D$_{test}$ | **75.3** | 74.6 | 72.3 | 66.3 |
| TextVQA$_{val}$ | 77.2 | **80.5** | 78.8 | 63.3 |
| *Math Reasoning* | | | | |
| MathVista$_{testmini}$ | **55.3** | 54.5 | 48.0 | 53.1 |
| *Hallucination* | | | | |
| POPE | **89.8** | 88.8 | 87.6 | 84.4 |

Table 4: Benchmarks performance of *Open-Qwen2VL* and other 2B-parameter state-of-the-art MLLMs.

We compare the final *Open-Qwen2VL* model with the state-of-the-art partially-open MLLMs of InternVL2.5-2B-MPO (Wang et al., 2024b), DeepSeekVL-2-Tiny (Wu et al., 2024), and Qwen2-VL-2B-Instruct (Wang et al., 2024a) on a set of general multimodal benchmarks, OCR VQA datasets, multimodal math reasoning benchmark, and hallucination benchmark. Concluded from the results in Table 4, *Open-Qwen2VL* demonstrates competitive performance across benchmarks compared to other 2B-parameter state-of-the-art MLLMs. It particularly excels in MMBench, achieving the highest score of 80.9, while maintaining comparable performance in SEEDBench and MMStar benchmarks. Moreover, *Open-Qwen2VL* outperforms the most relevant competitor Qwen2-VL-2B-Instruct on MMBench, SEEDBench-img, MMStar, and MathVista, while it is only trained on 0.35% of tokens of Qwen2-VL. However, it shows relatively weaker results in OCR VQA tasks of AI2D and TextVQA. This is because the pre-training data of *Open-Qwen2VL* does not include the OCR-specific caption dataset like SynthDoG (Kim et al., 2022) or LAIONCOCO-OCR (Schuhmann et al., 2022). Simply introducing such OCR-related pre-training data will significantly enhance the OCR-VQA task performance of MLLMs, as shown in Appendix Table 9.

## 4 Analysis

**Impacts of Sequence Packing on Multi-Image In-Context Learning and Reasoning.**
Flamingo (Alayrac et al., 2022) proposes MultiModal MassiveWeb (M3W) dataset to construct pseudo interleaving data structure using caption data to elicit the multimodal in-context learning capabilities of MLLMs. The multimodal sequence packing also constructs similar pseudo image-text interleaving data strcture. Thus, we conduct experiments to evaluate the few-shot multimodal in-context learning capabilities of the pre-trained base MLLM trained on packed multimodal sequences. We evaluate the base non-sft MLLM trained on the caption data mixture of CCS-CLIP and DataComp-MLM-Filter & DFN on GQA, VQA-v2, VizWiz, OKVQA, and Text-VQA datasets. This base model is the best model we get based on the ablation studies on the pre-training data mixture in Table 3. We select 5 random seeds for the 8-shot multimodal in-context learning experiments and report the average performance over the 5 random seeds. The results in Table 5 demonstrate that the base MLLM trained with packed multimodal sequences can learn from multimodal demonstration examples for completing the task well. The 8-shot in-context learning can gain +3% to +12% performance improvements on VQA dataset compared with 0-shot reasoning.

| # shots | GQA | VQA-v2 | VizWiz | OKVQA | Text-VQA |
|---------|------|--------|--------|-------|----------|
| 0 | 27.1 | 40.2 | 26.1 | 24.7 | 30.4 |
| 8 | 35.4 | 51.8 | 31.2 | 27.1 | 30.6 |

Table 5: Results of the 0-shot and 8-shot multimodal in-context learning capabilities of pre-trianed base MLLMs.

| Vision Encoder | AI2D test | TextVQA | POPE | MMMU val | MMBench Dev | SEEDBench Img-Dev | MMStar | MathVista test-mini | Avg. |
|----------------|-----------|---------|------|----------|-------------|-------------------|--------|---------------------|------|
| Frozen | 56.8 | 57.0 | 80.1 | 38.0 | 77.3 | 68.7 | 41.3 | 28.6 | 56.0 |
| Trainable | 57.4 | 57.6 | 82.3 | 36.1 | 76.5 | 69.7 | 41.4 | 29.3 | 56.3 |

Table 6: Ablation study on the trainable or frozen vision encoder during the SFT stage on LLaVA-665k data. We freeze vision encoder for pre-training stage due to limited computing resources.

This also demonstrates the necessity and significance of performing multimodal sequence packing as it can enhance the multimodal in-context learning capabilities of the MLLMs.

**Impact of Unfreezing Vision Encoder Parameters.** Most of the state-of-the-art MLLMs like InternVL-2.5 (Chen et al., 2024b) demonstrate that making the vision encoder trainable during the SFT stages will enhance the multimodal understanding capabilities of MLLMs. Therefore, we perform such ablation studies on our base non-sft MLLM pre-trained on CCS+DataComp-MLM-Filter&DFN data mixture. We use the LLaVA-665k data as the visual SFT dataset and evaluate the two SFT-ed MLLMs with frozen and trainable vision encoder during the SFT stage. The results in Table 6 demonstrates that make the vision encoder parameters trainable during SFT stage can achieve better average performance while there is significant performance degradation on MMMU benchmark.

**Advantage and Necessity of Large-Scale Efficient Pre-training.** Tn investigate the effects and necessity of performing large-scale efficient pre-training proposed in *Open-Qwen2VL*, we reproduce LLaVA using its original 595k pre-training caption data and 665k instruction data and then compare it with *Open-Qwen2VL*. The reproduced LLaVA adopts the architecture design and module choices and only differs with *Open-Qwen2VL* in the pre-training pipeline and data. Concluded from Table 7, our model significantly outperforms LLaVA-Reproduced on all benchmarks, demonstrating the significance and necessity of the large-scale efficient pre-training.

## 5 Related Work

**Open-Source Multimodal Large Language Models.** Close-source MLLMs like GPT-4o (Achiam et al., 2023) and Claude-3.7-Sonnet (Anthropic, 2025) have strong multimodal understanding and reasoning capabilities. To replicate the strong capabilities of close-source MLLMs, research teams from industry develop the open-weights strong MLLMs including InternVL-2.5 (Chen et al., 2024b), DeepSeek-VL2 (Wu et al., 2024), and Qwen2.5-VL (Bai et al., 2025), which can achieve equivalent capability with close-source models. However, the training data, codebase and data filtering details of such models are not open-sourced for reproduction. In additional to open-weights, the fully open-sourced MLLM including MOLMO (Deitke et al., 2024), SmolVLM (Marafioti et al., 2025), and our model open-source full training data and codebase to ensure the reproduction from scratch from the community.

**Large-Scale Image Text Data.** Starting from ImageNet (Deng et al., 2009), the large-scale image dataset has significantly driven advances in both computer vision and multimodal foundational models. MSCOCO (Lin et al., 2014), SBU (Ordonez et al., 2011), Conceptual Captions (CC) (Sharma et al., 2018b) scales up the image dataset size to near million level,

| Model | MMMU | MMBench | SEEDBench | MMStar | AI2D | Text-VQA | MathVista | POPE |
|---|---|---|---|---|---|---|---|---|
| LLaVA* | 35.3 | 73.8 | 65.3 | 38.5 | 56.0 | 49.9 | 24.7 | 75.7 |
| Open-Qwen2VL | **38.0** | **77.3** | **68.7** | **41.3** | **56.8** | **57.0** | **28.6** | **80.1** |

Table 7: Comparisons between *Open-Qwen2VL* and reproduced LLaVA on various multimodal benchmarks. Both *Open-Qwen2VL* and LLaVA adopts LLaVA-665k as the SFT data, while differing in pre-training pipeline and pre-training data.

which significantly enhances the image captioning performance of VLMs. OpenAI pretrained contrastive VLM, CLIP with 400M WebImage data without releasing them. Then LAION-400M and COYO-700M are open-source efforts to further scale up the image-text dataset to hundreds of million level. Then LAION-5B and DataComp-commonpool-12.8B scale up the image-text dataset size to billions level for supporting the data-intensive MLLM pre-training. Most of SOTA MLLMs like DeepSeek-VL (Wu et al., 2024), Qwen-VL (Bai et al., 2023), Intern-VL (Chen et al., 2023), SAIL (Dong et al., 2025) construct and curate their own large-scale image-text dataset with more then 10B image-text data, while such dataset will not be released for public research.

**High-Quality Image-Text Data Filtering.** Beyond the conventional rule-based or heuristic-based data filtering methods in constructing image-text dataset, image-text dataset with much larger scale for training contrastive VLMs adopts CLIPScore-based filtering methods for high-quality data curation. LAION-400M (Schuhmann et al., 2021) set a hard filtering threshold using OpenAI-CLIP model for its data filtering. Later, DataComp (Gadre et al., 2023) is the first effective benchmark for fairly evaluating the effectiveness of each data filtering method on selecting high-quality data for CLIP pre-training. Various methods (Maini et al., 2023; Yu et al., 2023; Wang et al., 2024d) try to combine CLIPScore filtering with other metrics to achieve better filtering performance on DataComp, while DFN (Fang et al., 2023) directly scales up the CLIP-based data filtering model and achieves the top-1 performance. Moreover, another line of data filtering method based on efficient MLLM (Wang et al., 2024c; Huang et al., 2024) emerges and presents better capabilities in selecting high-quality data for MLLM pre-training.

## 6 Conclusion

We demonstrate that the efficient MLLM-based high-quality data filtering techniques and well-designed data mixture strategies can achieve compute-efficient pretraining for developing SOTA MLLMs. Adopting the multimodal sequence packing and dynamic image token number with average-pooling layer can further enhance such pre-training efficiency. The induced final MLLM, *Open-Qwen2VL* outperforms the partially-open MLLM Qwen2-VL-2B on various multimodal benchmarks, in which *Open-Qwen2VL* is trained on only 0.36% pre-training tokens of Qwen2-VL. The training is conducted on academic-level computing resources and demonstrates that the advanced training pipeline and data filtering can overcome the limitation of computing resources. We wish *Open-Qwen2VL* can motivate the fully-open compute-efficient multimodal pre-training research from academic community.

## Acknowledgments

We would like to thank for Facebook (now Meta) for donating the 8xA100-40G GPUs for conducting the experiments. We appreciate the open-source contributions of prismatic-vlms Karamcheti et al. (2024) and vlm-evaluation[5], on which we build our training and evaluation codebases. This work was partially supported by the BioPACIFIC Materials Innovation Platform of the National Science Foundation under Award No. DMR-1933487.

---

[5]https://github.com/TRI-ML/vlm-evaluation

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

## A Training Settings of MLLM Pre-Training

The training details and hyperparameters for MLLM pre-training are presented in Tab. 8. The pre-training is only one-stage process. We do not follow Qwen-VL or DeepSeek-VL to split the MLLM pre-training into two stages of VL alignment and VL pre-training.

## B Effects of Additional OCR Data in Pre-Training

The deficiency of OCR performance of Open-Qwen2VL are mainly because the pre-training data of Open-Qwen2VL does not include any OCR caption data due to the pre-training efficiency considerations. We perform another ablation study and include the SynthDoG OCR (Kim et al., 2022) pre-training dataset, which introduces additional 0.13B tokens into the pre-training dataset. Then after SFT on LLaVA-665k, we observe that the model achieves significant improvements on OCR benchmarks of TextVQA and AI2D. It demonstrates that the deficiency of OCR performance is not caused by the methodology disadvantages but only the composition of pre-training data.

| Details | Pre-Training | Visual SFT |
|---|---|---|
| Vision Encoder | SigLIP-so400m-384px | SigLIP-so400m-384px |
| Visual Projector | 2d Adaptive Average Pooling + MLP | MLP |
| LLM Backbone | Qwen2.5-1.5B-Instruct | Qwen2.5-1.5B-Instruct |
| # Tokens per Image | 144 | 729 |
| Context Length | 4096 | 4096 |
| Sequence Packing | Yes | No |
| Precision | BF16 | BF16 |
| Global Batch Size | 256 | 128 |
| # Training Epoch | 1 | 1 |
| # GPUs | 8 A100-40G | 8 A100-40G |
| Peak LR | 5e-5 | 2e-5 |
| # Warmup Steps Ratio | 3% | 3% |
| LR Scheduler | linear-warmup+cosine-decay | linear-warmup+cosine-decay |
| Weight Decay | 0.01 | 0.1 |

Table 8: Training details and hyper-parameters for MLLM pre-training and visual supervised fine-tuning.

| Pre-Training Data | MMMU | MMBench | SEEDBench | MMStar | AI2D | Text-VQA | MathVista | POPE |
|---|---|---|---|---|---|---|---|---|
| CCS+DataComp | 38.0 | 77.3 | 68.7 | 41.3 | 56.8 | 57.0 | 28.6 | 80.1 |
| + SynthDoG | 37.9 | 77.0 | 69.6 | 41.0 | **57.6** | **59.7** | 28.8 | 82.0 |

Table 9: Ablation study on the effects of introducing additional OCR data in pre-training.

## C   Examples

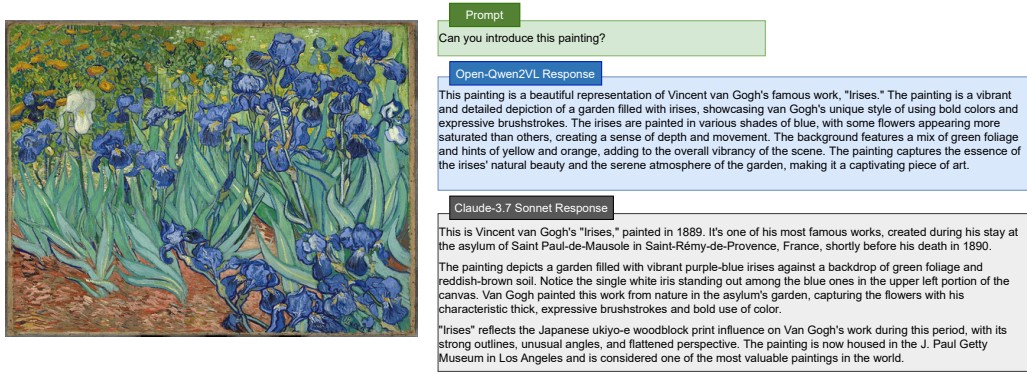

Figure 3: An example for generating a caption towards a famous painting, which requires the background knowledge.

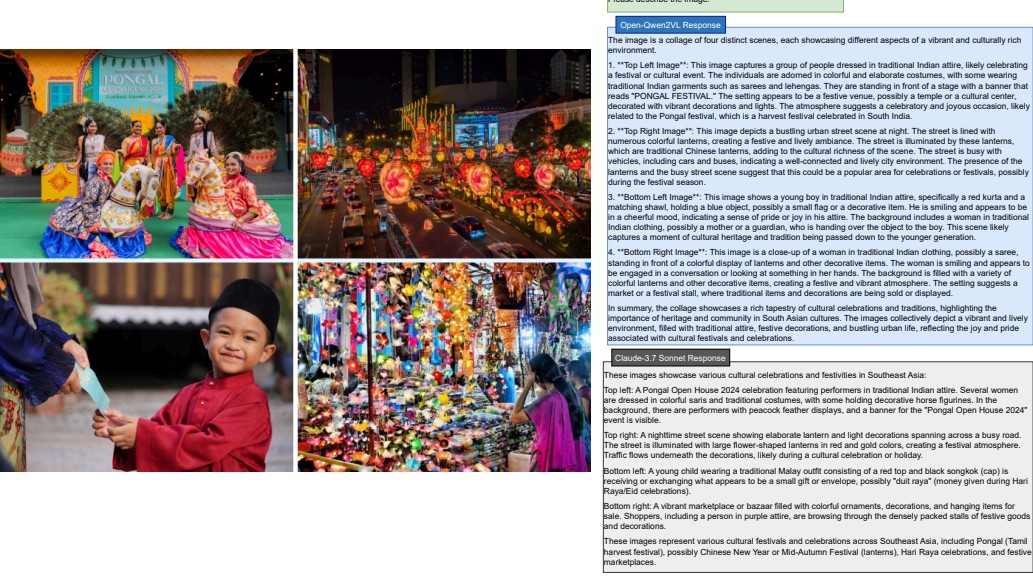

Figure 4: An example for generating long captions for a complicated 2x2 image.

