# OpenReview forum: "Open-Qwen2VL: Compute-Efficient Pre-Training of Fully-Open Multimodal LLMs on Academic Resources"
_colmweb.org/COLM/2025/Conference — COLM 2025_

### Official Review · Reviewer_G2dF · 2025-04-25

**Rating:** 7
**Confidence:** 3
**Ethics Flag:** 1

**Summary:**

This paper demonstrates that Multimodal LLMs can be effectively trained with academic-level resources. Specifically, it presents Open-Qwen2VL, a 2B-parameter MLLM trained with only 442 A100-40G GPU hours. The model was pre-trained on 5B tokens, which is just 0.36% of the 1.4T tokens used for Qwen2-VL. Despite the much smaller compute and data, Open-Qwen2VL achieves comparable or even better performance compared to Qwen2-VL.
To enable compute-efficient training, the authors introduce techniques like low-to-high dynamic image resolution and multimodal sequence packing. The training dataset was also carefully curated using various filtering methods.
Importantly, this paper fully open-sources the entire training pipeline, including the model, data, code, and filtering scripts, contributing significantly to the research community by making MLLM pretraining accessible to those with limited resources.

**Questions To Authors:**

Please refer to the Reasons to Reject section above.

**Reasons To Accept:**

1. The paper demonstrates that it is possible to train a strong MLLM using only academic-level resources.
2. It effectively leverages techniques such as low-to-high dynamic resolution to maximize cost-efficiency, highlighting strategies that are often overlooked by industry models with abundant compute.
3. It makes a major contribution to the community by fully open-sourcing not just the checkpoints but all the components necessary for training.

**Reasons To Reject:**

1. The paper does not provide a comparison with concurrent works targeting similar goals, such as Molmo [1] and SmolVLM [2]. A detailed analysis comparing Open-Qwen2VL against these models in terms of training cost, model performance, and degree of open-sourcing for reproducibility would substantially strengthen the paper.
2. Models like LLaVA, which do not rely on multimodal pretraining, have demonstrated that strong MLLMs can be trained using only projector training and visual instruction tuning with academic-level resources. In this context, it is important for the paper to justify why full multimodal pretraining is necessary under similar resource constraints, and to show whether it provides meaningful advantages over visual instruction tuning alone.
3.  The paper does not explicitly evaluate the model on multi-image or video tasks. It remains unclear whether Open-Qwen2VL achieves comparable performance to Qwen2-VL in these settings.

[1] Molmo and PixMo: Open Weights and Open Data for State-of-the-Art Vision-Language Models
[2] SmolVLM: Redefining small and efficient multimodal models

---

> ### Author Response · Authors · 2025-06-02
>
> Thank you for your comprehensive reviews and helpful suggestions.
>
> **Question 1: Comparisons with Molmo and SmolVLM**
>
> Molmo is not comparable with Open-Qwen2VL-2B since the model series of Molmo does not have the comparable model size of 2B. We compare our models with SmolVLM in the following table (The results of SmolVLM are cited from their system card):
>
> | Model | MMMU | MMBench | SEEDBench | MMStar | AI2D | Text-VQA | MathVista | POPE |
> |---------------------|------|---------|-----------|----------|------|---------|-----------|----------|
> | SmolVLM | 38.8 | - | - | 46.6 | - | 72.7 | 44.6 | - |
> | Open-Qwen2VL-2B | 39.8 | 80.9 | 72.5 | 49.7 |  66.3|  63.3 | 53.1 |  84.4 |
>
> Moreover, our model focuses most on the efficient pre-training and the efficient model architecture design while SmolVLM only has the SFT stage. SmolVLM adopts a slightly larger SFT dataset with about 12M+ instruction data compared with ours. Secondly, Molmo-1B model takes 2k H100 GPU hours, equivalent to 6k A100-40g hours, which is 15x times of our pre-training cost used for training a 2B model.
>
> Both Molmo/SmolVLM and ours open-source the code, data, and checkpoints. Molmo does not provide enough details for the curation and construction of their pre-training data PixMo-Cap.
>
>
>
>
> **Question 2: Advantage and Necessity of Large-Scale Efficient Pre-training**
>
>
> We reproduce LLaVA based on Qwen-1.5B model, which is used in our Open-Qwen2VL model, using the original efficient 595k pre-training caption data and 665k instruction data. Then we compare such LLaVA without efficient large-scale pre-training with Open-Qwen2VL in the following table. Our model significantly outperform LLaVA-Qwen2 with significant improvements on all benchmarks, demonstrating the significance and necessity of the large-scale efficient pre-training.
>
> | Model | SFT-Data |  MMMU | MMBench | SEEDBench | MMStar | AI2D | Text-VQA | MathVista | POPE |
> |---------------------|------|---------|-----------|----------|---------------------|------|---------|-----------|----------|
> | LLaVA-Qwen2  | LLaVA-665k | 35.3 | 73.8 | 65.3 | 38.5 | 56.0 | 49.9 | 24.7 | 75.7 |
> | Open-Qwen2VL-2B | LLaVA-665k | 38.0 | 77.3 | 68.7 | 41.3 | 56.8 | 57.0 | 28.6 | 80.1 |
>
> **Question 3: Multi-image benchmarks**
> We evaluate both the Open-Qwen2VL and Qwen2-VL on the multi-image understanding benchmark, Mantis-Eval [1]. The results in the following table demonstrates that the Open-Qwen2VL achieves slightly better performance than Qwen2-VL on multi-image understanding benchmarks. Since the Open-Qwen2VL is not pre-trained on video data, we did not eval it on video benchmarks. We will include the LLaVA-One-Vision instruction data into the SFT stage of Open-Qwen2VL to strengthen its video understanding capabilities and perform evaluation on video benchmarks in future versions.
>
>
> | Model | Mantis-Eval |
> |---------------------|------|
> | Qwen2VL | 46.1 |
> | Open-Qwen2VL | 46.5 |
>
>
> [1] Jiang et al. Mantis: Interleaved multi-image instruction tuning.

---

> > ### Comment · Reviewer_G2dF · 2025-06-03
> >
> > Thank you for the response. I have raised my score from 6 to 7.

---

### Official Review · Reviewer_eK8t · 2025-05-05

**Rating:** 7
**Confidence:** 3
**Ethics Flag:** 1

**Summary:**

The paper presents Open-Qwen2VL, a fully open-source 2B-parameter multi-modal LLM pre-trained on just 5B tokens. The authors rely on careful data curation using both MLM-based filtering techniques and CLIP-based filtering methods, and increase training efficiency by packing tokens into sequences and leveraging low-to-high dynamic image resolution. The resulting model outperforms Qwen2-VL on select benchmarks. The authors promise to release the model along with the full training data and training code.

**Questions To Authors:**

Please respond to the questions raised in "Weaknesses"

**Reasons To Accept:**

Demonstrates how to increase pre-training efficiency,  which makes such research more accessible. Promised open release of training artifacts will be valuable for the community.

**Reasons To Reject:**

The reported benchmarks in Table 4 are a substantially smaller list than e.g. Table 2 in Qwen2, leaving open how the model performs on the remaining tasks. That raises the question whether the claimed training efficiency might come from improving on some (the reported) tasks at the cost of other (non-reported) tasks. The results and related discussion regarding OCR tasks also hint in this direction.

---

> ### Author Response · Authors · 2025-06-02
>
> Thank you for your helpful reviews and suggestions.
>
>
> **Question 1: Broader Evaluations**
>
> Firstly, we set up a very comprehensive benchmark set in the original evaluation settings based on our experience in robust MLLM evaluation. Compared with Table 2 in Qwen2, we include additional milestone benchmarks of SeedBench and POPE into the evaluation set and cover the most important benchmarks of MMMU, MMBench, MathVista, and MMStar to ensure that all models are evaluated and compared in fair and comprehensive way. Some evaluation benchmarks in Table 2 of Qwen2 like VCR and MTVQA are not commonly-used benchmarks in MLLM evaluations.
>
> The benchmark set used in our work can ensure the comprehensive evaluations of the capabilities of MLLM across general knowledge reasoning, OCR, visual question answering, visual math reasoning, and hallucination. In this way, we can conclude that the Open-Qwen2VL achieves robust and comprehensive capability improvements rather than the overfitting to a small set of capabilities or benchmarks.
>
> We do agree that including more and more evaluation benchmarks can always help observe the behavior and capabilities of models. Thus, we make additional engineering efforts to include another significant benchmark RealWorldQA [1] used in Qwen2 Table2 into our evaluation set and evaluate the two models with our eval codebase.
>
>
> | Model | RealWorldQA |
> | ------- | -------- |
> | Qwen2-VL-2B | 60.9 |
> | Open-Qwen2VL-2B | 62.3 |
>
> **Question 2: OCR Performance**
>
>
> We explained in the paper about the deficiency of OCR performance of Open-Qwen2VL because the pre-training data of Open-Qwen2VL does not include any OCR caption data due to the pre-training efficiency considerations. We perform another ablation study and include the SynthDoG OCR pre-training dataset, which is about another 0.13B tokens into the pre-training dataset. Then after SFT on LLaVA-665k, we observe that the model achieves significant improvement on OCR benchmarks of TextVQA and AI2D. The deficiency of OCR performance is not caused by any methodology disadvantages but only the composition of pre-training data.
>
> | Pre-Training Data | SFT Data | MMMU | MMBench | SEEDBench | MMStar | AI2D | Text-VQA | MathVista | POPE |
> |---------------------|------|---------|-----------|----------|---------------------|------|---------|-----------|----------|
> | CCS + DataComp | LLaVA-665k | 38.0 | 77.3 | 68.7 | 41.3 | 56.8 | 57.0 | 28.6 | 80.1 |
> | CCS + DataComp + SynthDoG |  LLaVA-665k | 37.9 | 77.0 | 69.6 | 41.0 | **57.6**  |  **59.7** |  28.8 |  82.0 |
>
>
> [1] RealWorldQA. xAI. Grok-1.5 vision preview. 2024.

---

> > ### Comment · Reviewer_eK8t · 2025-06-04
> >
> > Thank you. The response has mitigated my concerns, I will adjust my score.

---

### Official Review · Reviewer_RPfn · 2025-05-11

**Rating:** 7
**Confidence:** 3
**Ethics Flag:** 1

**Summary:**

The paper introduces Open-Qwen2VL, a 2B-parameter multimodal large language model (MLLM) trained with only 442 A100-40G GPU hours on 29M curated image-text pairs. It emphasizes full openness of data, code, and filtering pipelines, and demonstrates competitive performance against state-of-the-art models using significantly fewer resources. Key contributions include efficient data filtering using MLLMs, multimodal sequence packing, and a low-to-high visual resolution scheme. The final instruction-tuned model surpasses Qwen2-VL-2B in several benchmarks, while being more resource-efficient.

**Reasons To Accept:**

All components of the project—including the training dataset, filtering pipeline, codebase, and final model checkpoints—are released, which significantly contributes to the transparency and usability of multimodal model research.

The model achieves competitive or even superior performance while using a fraction of the training budget required by similar models, making it highly relevant to resource-constrained settings.

Techniques such as sequence packing and low-to-high image resolution scheduling are well-motivated and demonstrably improve training efficiency without architectural modifications.

**Reasons To Reject:**

The backbone and visual encoder are largely reused; the work focuses on data and training improvements rather than introducing new model designs or training objectives.

Some engineering details lack theoretical justification, and could benefit from more systematic analysis.

While the ablations are helpful, the work would benefit from additional analysis, such as statistical significance tests, longer training runs, or cross-dataset generalization evaluation.

---

> ### Author Response · Authors · 2025-06-02
>
> Thank you so much for your helpful suggestions on improving the overall quality of our work.
>
> **Question 1: Architecture Design Contributions**
>
>
> We also contributed to model architecture design. We propose to adopt the most efficient and non-parametric Average-Pooling projector and meanwhile propose the low-to-high dynamic resolution design to enhance the model efficiency.
>
>
> **Question 2: Additional Analysis**
>
> We perform a statistical significance test on the MMBench benchmark between Open-Qwen2VL-2B and Qwen2VL-2B.
>
> |                        | Model 2 Correct | Model 2 Wrong |
> |---------------------|------|---------|
> | Model 1 Correct |     3106      |      248    |
> | Model 1 Wrong   |      434      |      589    |
>
> chi^2=248.0, p=1.02e-12<0.05. Therefore, The conclusion is that the difference in performance between the models is statistically significant and Open-Qwen2VL performs significantly better than Qwen2VL on MMBench.
>
> **Question 3: Longer training run and more datasets**
>
> We perform another ablation study and include the SynthDoG OCR [1] pre-training dataset, which is about another 0.13B tokens into the pre-training dataset. Then after SFT on LLaVA-665k, we observe that the model achieves significant improvement on OCR benchmarks of TextVQA and AI2D.
>
> | Pre-Training Data | SFT Data | MMMU | MMBench | SEEDBench | MMStar | AI2D | Text-VQA | MathVista | POPE |
> |---------------------|------|---------|-----------|----------|---------------------|------|---------|-----------|----------|
> | CCS + DataComp | LLaVA-665k | 38.0 | 77.3 | 68.7 | 41.3 | 56.8 | 57.0 | 28.6 | 80.1 |
> | CCS + DataComp + SynthDoG |  LLaVA-665k | 37.9 | 77.0 | 69.6 | 41.0 | **57.6**  |  **59.7** |  28.8 |  82.0 |
>
>
> [1] Kim et al. OCR-Free Document Understanding Transformer.

---

> > ### Comment · Reviewer_RPfn · 2025-06-07
> >
> > Appreciate your effort. My concerns are addressed.

---

### Official Review · Reviewer_aZwW · 2025-05-12

**Rating:** 7
**Confidence:** 4
**Ethics Flag:** 1

**Summary:**

The paper introduces Open-Qwen2VL, a 2B-parameter multimodal LLM pre-trained efficiently on 29M image-text pairs using academic-grade resources (8×A100-40G GPUs, 442 GPU hours). Key contributions include dynamic image resolution scaling (144→729 tokens), multimodal sequence packing to reduce padding, and hybrid data filtering (CLIP + MLM-Filter). The model outperforms Qwen2-VL-2B on benchmarks (MMBench, SEEDBench) despite using 0.36% of its pre-training tokens. The authors emphasize full openness, releasing code, data, and methodologies.

The paper is well written and makes a huge impact for the academia. The drawbacks like OCR underperformance, are mentioned in the limitations. The only point I want to outline - the more comparisons make the paper more strong.

**Questions To Authors:**

Why exclude interleaved image-text data during pre-training despite its potential for in-context learning, as seen in Flamingo?
What motivation is to use of SigLIP-SO-400M over other vision encoders, and how does it impact training efficiency?

**Reasons To Accept:**

1. Achieves SOTA performance with minimal resources,
2. The release of the training codebase and pre-training/SFT data democratizes MLLM development for academic researchers.
3. Low-to-high resolution (144→729 tokens) and sequence packing improve training efficiency without sacrificing quality.

**Reasons To Reject:**

1. Incomplete OCR Analysis. While TextVQA/AI2D performance lags (Table 4), the paper does not explore integrating OCR-specific pre-training data (e.g., SynthDoG).
2. Only 4 of 16 possible data combinations tested (Table 3), leaving optimal mixtures understudied
3. The impact of freezing the vision encoder during pre-training vs. SFT is underexplored.
4. Limited comparisons with other open-source MLLMs (e.g., LLaVA, OpenFlamingo)

---

> ### Author Response · Authors · 2025-06-02
>
> Thank you for the helpful review and suggestions.
>
> **Question 1: Including OCR Pre-training data**
>
> We perform another ablation study and include the SynthDoG OCR pre-training dataset, which is about another 0.13B tokens into the pre-training dataset. Then after SFT on LLaVA-665k, we observe that the model achieves significant improvement on OCR benchmarks of TextVQA and AI2D.
>
> | Pre-Training Data | SFT Data | MMMU | MMBench | SEEDBench | MMStar | AI2D | Text-VQA | MathVista | POPE |
> |---------------------|------|---------|-----------|----------|---------------------|------|---------|-----------|----------|
> | CCS + DataComp | LLaVA-665k |                      38.0 | 77.3 | 68.7 | 41.3 | 56.8 | 57.0 | 28.6 | 80.1 |
> | CCS + DataComp + SynthDoG |  LLaVA-665k | 37.9 | 77.0 | 69.6 | 41.0 | **57.6**  |  **59.7** |  28.8 |  82.0 |
>
>
>
>
> **Question 2: Data Mixture Ablation Settings**
>
> The search space of the data mixture is too large which may take unacceptable computation costs. CCS is on a million level scale and regarded as high-quality data in early practices, i.e. BLIP-1 pre-training. Thus, we reserve CCS in all data mixture configurations and decrease the 16 configurations into 8. Secondly, the DataComp and LAION are both constructed by web-crawl data and have huge distribution overlap. Via comparing the CCS+DataComp and CCS+LAION data mixture configurations, CCS+DataComp outperforms the mixture of CCS+LAION. Then, we also find that combining both DataComp and LAION did not improve the performance because the data diversity and quality may not be improved via simple concatenating. Therefore, we introduce another data filtering model MLMFilter to work with DFN filters to curate high-quality data from DataComp and achieve the best performance. Therefore, the 4 ablated data mixture configurations are comprehensive and cover the most potential data mixture strategies.
>
>
>
> **Question 3: Effects of Trainable Vision Encoder**
>
> We perform the ablation study on the effects of trainable vision encoder in Table 6 and find that trainable vision encoder yields slightly better performance. According to the recent advances in MLLM pre-training like Qwen2VL and Intern2.5VL, making the vision encoder parameters trainable will not bring significant benefit but introduce quite huge computational overhead. Following our research goal of efficient training of MLLMs in this work, we freeze the vision encoder all the time during pre-training and SFT to save computational cost.
>
> **Question 4: Comparisons with other open-source models**
>
> We compare our 2B models with larger open-source models of LLaVA-1.5-13B and Open-Flamingo-9B and find that our models significantly outperform them on the multimodal benchmarks. These two models are developed in 2023 and do not maintain the SOTA performance in 2025. This is why we mainly compare our model with Intern2.5VL and Qwen2VL in the Table 4.
>
>
> | Model               | MMMU | MMBench | SEEDBench | Text-VQA |
> |---------------------|------|---------|-----------|----------|
> | LLaVA-1.5-13B | 36.4 | 67.7 | 61.6 | 61.5 |
> | Open-Flamingo-9B |  28.7 | 36.7 | 33.1 | 24.2 |
> | Open-Qwen2VL-2B | **39.8** | **80.9** |  **72.5** | **63.3** |
>
> **Question 5: Interleaved Data**
>
> The interleaved data is vital for improving the in-context learning capabilities. But our computational resources are limited and we only include the caption data in pre-training. If we can find more computational resources, we will consider including the OBELICS interleaved dataset to enhance the multimodal in-context learning capabilities of the model.
>
> **Question 6: Choice of Vision Encoder**
>
> The wide usage of SigLIP vision encoder instead of CLIP vision encoder starts with Prismatic-VLM [1] and Cambrian-1 [2]. Based on their ablation studies, adopting SigLIP vision encoder can significantly outperform CLIP. SigLIP will introduce 729 (27*27) tokens for one image while CLIP will use 576 (24*24). But our proposed average-pooling visual projector can downscale the 729 output patches into 144 visual tokens to represent one image, which resolves the issue of additional computational overhead.
>
> [1] Prismatic VLMs: Investigating the Design Space of Visually-Conditioned Language Models.
>
> [2] Cambrian-1: A Fully Open, Vision-Centric Exploration of Multimodal LLMs

---

> > ### Comment · Reviewer_aZwW · 2025-06-07
> >
> > Thank you for your thorough explanation. Your response has successfully addressed my previous concerns, and based on this clarification, I will adjust my evaluation score accordingly.

---

### Decision · Program_Chairs · 2025-07-08

**Decision:**

Accept

**Comment:**

This paper introduces Open-Qwen2VL,  a compact 2B-parameter multimodal model trained with minimal compute, emphasizing efficiency and full transparency. The authors leverage techniques like dynamic resolution scaling, sequence packing, and curated data filtering to reduce resource requirements while maintaining strong performance. The model achieves competitive results across diverse benchmarks and demonstrates meaningful improvements over comparable models, such as Qwen2VL, LLaVA, SmolVLM, etc. These results demonstrate the effectiveness of the proposed training recipe to attain strong yet cost-efficient VLM. The full release of code, data, and training pipelines will significantly enhance reproducibility and accessibility for academic research.

During rebuttal, the authors further effectively addressed reviewer concerns through additional experiments, ablations, and broader comparisons. As such, all reviewers reached a consensus about the contributions of this work and all gave a high rating (7, accept) on this work. The Area Chairs agreed with all reviewers and appreciate the authors' work and rebuttal, and recommend a clear acceptance.